# Predicting Hearing Loss in Testicular Cancer Patients after Cisplatin-Based Chemotherapy

**DOI:** 10.3390/cancers15153923

**Published:** 2023-08-01

**Authors:** Sara L. Garcia, Jakob Lauritsen, Bernadette K. Christiansen, Ida F. Hansen, Mikkel Bandak, Marlene D. Dalgaard, Gedske Daugaard, Ramneek Gupta

**Affiliations:** 1Department of Health Technology, Technical University of Denmark, 2800 Kongens Lyngby, Denmark; sara.blg07@gmail.com (S.L.G.); ramg@dtu.dk (R.G.); 2Novo Nordisk Foundation Center for Protein Research, Faculty of Health and Medical Sciences, University of Copenhagen, 1958 Copenhagen, Denmark; 3Department of Oncology, Copenhagen University Hospital, 2730 Copenhagen, Denmark; 4Department of Computational Biology, Novo Nordisk Research Centre Oxford, Oxford OX3 7FZ, UK

**Keywords:** chemotherapy regimen, genetics, testicular cancer, hearing loss, machine learning

## Abstract

**Simple Summary:**

To our knowledge, this is the first study that presents a machine learning setup incorporating genetics and clinical factors to predict hearing loss in a large and fairly unique cohort of testicular cancer patients, with follow-up data examining long-range side effects of chemotherapy. Genetic variants in *SOD2* and *MGST3* are proposed as mechanistically associated with cisplatin-induced hearing loss. Further, the models in this study focus on individual patient benefit and incorporation of quality of life measures to identify hearing loss impact. To study short- and long-term effects of chemotherapy, testicular cancer is ideal as a model disease for other cancers, as patients are young with long life-expectancy and without significant comorbidity. With small adjustments, the model can likely be applied in the treatment of other cancers where cisplatin is used, thus helping with choice of treatment without risking a trade-off in efficacy, standing to influence clinical practice.

**Abstract:**

Testicular cancer is predominantly curable, but the long-term side effects of chemotherapy have a severe impact on life quality. In this research study, we focus on hearing loss as a part of overall chemotherapy-induced ototoxicity. This is a unique approach where we combine clinical data from the acclaimed nationwide Danish Testicular Cancer (DaTeCa)-Late database. Clinical and genetic data on 433 patients were collected from hospital files in October 2014. Hearing loss was classified according to the FACT/GOG-Ntx-11 version 4 self-reported Ntx6. Machine learning models combining a genome-wide association study within a nested cross-validated logistic regression were applied to identify patients at high risk of hearing loss. The model comprising clinical and genetic data identified 67% of the patients with hearing loss; however, this was with a false discovery rate of 49%. For the non-affected patients, the model identified 66% of the patients with a false omission rate of 19%. An area under the receiver operating characteristic (ROC-AUC) curve of 0.73 (95% CI, 0.71–0.74) was obtained, and the model suggests genes SOD2 and MGST3 as important in improving prediction over the clinical-only model with a ROC-AUC of 0.66 (95% CI, 0.65–0.66). Such prediction models may be used to allow earlier detection and prevention of hearing loss. We suggest a possible biological mechanism for cisplatin-induced hearing loss development. On confirmation in larger studies, such models can help balance treatment in clinical practice.

## 1. Introduction

Testicular cancer is the most common cancer in men below 40 years of age in developed countries with a continuously rising incidence in many countries [1]. It is a highly curable disease with a 5-year survival of more than 90% disregarding initial stage, which results in an increasing population of long-term testicular cancer survivors (TCS) [2,3]. Treatment for patients with disseminated disease includes a multi-modality approach with initial orchiectomy and either radiotherapy or, most frequently, chemotherapy, followed by possible secondary surgery. Standard chemotherapy is bleomycin–etoposide–cisplatin, which has been the standard of care since the early 1980s [4] in generally unchanged regimens. While treatment is associated with high cure rates, it is hampered by late effects such as ototoxicity, neurotoxicity, nephrotoxicity, and cardiovascular disease [5,6].

Prevalence of ototoxicity has been associated with cumulative cisplatin doses and age at diagnosis. For example, with every 100 mg/m^2^ increase in cisplatin dose, a 3.2 dB decrease in age-adjusted hearing threshold has been identified [7]. In a large cross-sectional study, hearing impairment was reported in 21% of patients who received chemotherapy compared to 12% in those who did not [8]. In another cross-sectional study, 18.3 years after treatment, treatment with BEP was associated with 2.4–2.8 times increased risk of self-reported ototoxicity [9]. There is, however, considerable interindividual variability, which may be caused by genetic factors [9,10,11].

Platinum ototoxicity has been associated with death of hair cells in the organ of Corti, which reside within a blood–labyrinth barrier [10,12], which may hold a key to understanding the variability of toxicity. There is a need for further identification of risk factors to more accurately ascertain which patients are at risk of ototoxicity and possibly initiate preventive measures. In this study, we aimed to identify risk factors for hearing loss after cisplatin-based chemotherapy, as part of ototoxicity, in TCS via the usage of a prediction logistic regression model integrating clinical and genetic data to address the burden of cisplatin-induced hearing loss.

## 2. Materials and Methods

### 2.1. Source of the Data

Long-term TCS were identified in the DaTeCa-Late cohort [13] with patients initially treated for testicular cancer in Denmark from 1984 to 2007. All patients in this cohort filled in a range of questionnaires related to late toxicity from January 2014 to December 2016 (*n* = 2572). Clinical features were identified in hospital files [14]. In October 2014, 433 of these TCS who had received one line of treatment were asked to deliver a saliva sample for genotyping, as previously described [15].

Patients gave informed consent to participate in this study, and the study was approved by the regional ethical committee (File number, H-2-2012-044) and the National Board of Data Protection (File number, 2012-41-0751).

### 2.2. Treatment and Clinical Information

All patients received bleomycin–etoposide–cisplatin (BEP) for disseminated testicular cancer, for three cycles or more, as previously described [15]. The majority received cisplatin 20 mg/m^2^ and etoposide 100 mg/m^2^ for five days each cycle, while 43 (10%) received double-dose cisplatin (40 mg/m^2^) and etoposide (200 mg/m^2^) as part of a research protocol. Bleomycin was administered at a dose of 15.000 IE/m^2^ with a cumulative maximum dose of 150.000/m^2^.

Clinical information consisted of age at diagnosis and at time of completion of the questionnaire, body mass index (BMI), glomerular filtration rate before treatment, cumulative cisplatin dose per square meter of body surface area (BSA), number of BEP cycles, histology (seminoma vs. non-seminoma), prognostic classification as per International Germ Cell Cancer Collaborative Group (IGCCCG) [16], alcohol consumption (units/week), and smoking habits (never; former; or current). BMI, alcohol, and smoking information were collected at the time of the questionnaire. Age at time of completion of the questionnaire was correlated with age at diagnosis (Pearson correlation 0.76) and omitted for further analysis.

### 2.3. Assessment of Hearing Loss

Self-perceived hearing loss was assessed with the Ntx subscale of the FACT/GOG-Ntx-11, version 4, which evaluates the severity and impact of neuropathy [17]. The questionnaire consists of 11 items rated from 0 (not at all) to 4 (very much). The scale can be divided into four subscales: sensory neuropathy, motor neuropathy, auditory neuropathy, and dysfunctional problems [17]. Auditory neuropathy comprises two different questions, where FACT/GOG-Ntx6 measures difficulty hearing, and FACT/GOG-Ntx7 measures tinnitus (Appendix A).

Here, we aim at predicting hearing loss; thus, only FACT/GOG-Ntx6 is further explored. FACT/GOG-Ntx6 and FACT/GOG-Ntx7 were not strongly correlated, which may indicate different biological etiologies. For FACT/GOG-Ntx6, to ensure clinical relevance, the outcome was dichotomized. Low-risk (score from 0 to 1) and high-risk groups (score from 2 to 4) were considered.

It is important to point out that the FACT/GOG-Ntx questionnaire was completed in 2014, and patients had answered FACT/GOG-Ntx6 according to their current experience of hearing levels. However, at that time, the patients were also asked if they recalled experiencing worse hearing during treatment (hearing change question 1, HC Q1), and whether it returned to normal afterwards (hearing change question 2, HC Q2). Even though HC Q1 and HC Q2 are not validated at the same level as FACT/GOG-Ntx [17], we used Spearman’s rank correlation between FACT/GOG-Ntx6 and HC Q1 and HC Q2 to understand if the reported patients’ hearing loss at the time of the FACT/GOG-Ntx questionnaire was due to cisplatin treatment.

### 2.4. DNA Preparation and Quality Control

DNA samples were prepared at DTU Multi-Assay Core (Lyngby, Denmark) and genotyped at AROS Applied Biotechnology A/S company (Aarhus, Denmark) using Illumina^®^ HumanOmniExpressExome-8-v1-2-B-b37 chip (approximately 1 million markers).

Genotyping data were converted into pedigree format using GenomeStudio^®^ (v2011.1) with PLINK Input Report Plug-in (v2.1.3). Variants with genotyping call rate < 0.98, not in Hardy–Weinberg equilibrium (*p* value < 5 × 10^−6^), or with a minor allele frequency < 0.01 were excluded. Quality control for both single nucleotide polymorphisms (SNPs) and patient samples is described in detail in Appendix A.

### 2.5. Genetic Data Feature Selection

SNPs were selected via a genome-wide association study (GWAS) within a nested cross-validated logistic regression (described in “Statistical Analysis”) and a systematic review search. Genes linked with cisplatin metabolism or ototoxicity were obtained (Table 1 and Appendix A) from databases Uniprot [18], DrugBank [19], KEGG [20,21,22], and BioCyc [23]. SNPs from our dataset were gene-annotated with Ensembl Variant Effect Predictor (VEP) [24], and SNPs located in the database-derived prior genes were extracted. SNPs were further filtered using Ensembl VEP for high functional impact or drug response, thus prioritizing 19 SNPs representing prior knowledge: *CYP2J2* rs11572279, *MGST3* rs9333378, *ABCA12* rs10498027, *ABCC5* rs939336, *WFS1* rs1801206, *SLC44A4* rs494620, *NOX3* rs12195525, *CEP78* rs17787781, *CYP2C9* rs4917639, *CYP2C8* rs2071426, *SYCE1* rs2149616, *ABCC8* rs2074308, *DUSP6* rs808820, *DMXL2* rs2414105, *ABCA10* rs10491178, *ABCA7* rs3752229, *CYP2B6* rs2279345, *ERCC1* rs3212986, *MCM8* rs3761873. Additionally, seven SNPs reported to be associated with cisplatin-induced hearing loss in a recent systematic review [25] were included: *LRP2* rs2075252, *LRP2* rs4668123, *TPMT* rs1800460, *SOD2* rs4880, *GSTP1* rs1695, *COMT* rs4646316, *COMT* rs9332377.

### 2.6. Statistical Analysis

#### 2.6.1. Missing Data

All patients included after quality control had complete hearing loss outcome data. In patients with missing values in predictors (for BMI and smoking), a multiple imputation method [26,27] with ten iterations was used.

#### 2.6.2. Logistic Regression with Cross-Validated GWAS

A nested five-outer, five-inner cross-validation logistic regression was implemented using SciKit-learn [28] (v0.23.2) in Python (v3.6.10). As performance was similar across other machine learning models (random forests and artificial neural networks), the more simplistic logistic regression was chosen to be closest to interpretability and eventual implementation.

Forward feature selection and parameter optimization were performed in the inner training-validation sets, and the model was deployed on the outer test sets. Initially, only clinical data were included in the model. The area under the ROC-AUC was used to evaluate the model’s prediction ability. An increasing number of clinical features was evaluated in exhaustive combinations until the ROC-AUC reached a plateau. 

The genetic data were then added to the model. A cross-validated GWAS was performed on the inner training sets to select SNPs for model training. Genetic variants were tested for association with hearing loss using logistic regression (PLINK [29] (v1.9)) adjusting for potential confounding effects: age at time of questionnaire and cisplatin dose. A suggestive *p* value threshold of 1 × 10^−4^ was used to select SNPs for model training (Appendix A). Then, forward feature selection was performed on the combined dataset comprising both SNPs identified through GWAS and a systematic review search, along with the clinical data. SHapley Additive exPlanations (SHAP) values [30] helped interpret the impact of individual features contributing to the model’s performance.

The dataset was randomly split 30 different times in training, validation, and test sets to ensure model reproducibility and robustness. More information on model hyperparameters, encoding of variables, and feature normalization is included in Appendix A.

For the model with clinical data only, permutation tests were applied to ensure the model was not fitting random noise. For the model with clinical and genetic data, this was achieved by adding randomly selected SNPs.

## 3. Results

Out of 478 patients from the Danish Testicular Cancer (DaTeCa)-Late cohort [13], 45 patients who received more than one line of treatment were excluded in the present study; therefore, 433 patients were available. Out of these 433 patients, 424 filled in the FACT/GOG-Ntx6 question on self-perceived hearing loss. Of those, 146 (34.4%) patients scored 2 to 4, phenotypical hearing loss. These affected patients had a median age at diagnosis (interquartile range (IQR)) of 34 (27–41) years, while non-affected (*n* = 278) patients had a median age (IQR) of 29 (26–36) years. Demographic features are presented in Table 2.

After genotype quality control, 393 patients with data on 611,129 SNPs were available for analysis (Appendix A).

In this study, we used the FACT/GOG-Ntx-11 version 4, which provides a targeted assessment of peripheral neuropathy such as auditory neuropathy. Auditory neuropathy comprises two different questions, where FACT/GOG-Ntx6 measures difficulty hearing, and FACT/GOG-Ntx7 measures tinnitus (Appendix A). A moderate correlation was observed between FACT/GOG-Ntx6 and FACT/GOG-Ntx7 (Spearman’s rank correlation coefficient 0.55). Additionally, the patients were asked if they recalled experiencing worse hearing during treatment (hearing change question 1, HC Q1) and whether it returned to normal afterwards (hearing change question 2, HC Q2). In order to understand if the patient’s hearing loss at the time of the FACT/GOG-Ntx questionnaire (2014) was due to cisplatin treatment (between 1984 and 2007), we investigated the correlation between these questions as well. FACT/GOG-Ntx6 from the validated FACT/GOG-Ntx questionnaire showed a high correlation with HC Q2 concerning self-perceived long-lasting changes after treatment (Spearman’s rank correlation coefficient 0.56 for HC Q1 and 0.76 for HC Q2).

First, the prediction ability of the routinely available clinical information was assessed. Nine features were incrementally included in the model through exploring exhaustive permutations with each single feature addition. The two most informative clinical features (receiver operating characteristic curve ROC-AUC of 0.66 (95% CI, 0.65–0.66), Figure 1A,C)—age at diagnosis and number of treatment cycles—were prioritized for further modeling and combined with genetic data from the SNP array chip.

Prediction performance, assessed as ROC-AUC, reached a plateau when six genetic features were added to the model (in addition to the two most informative clinical parameters), with a mean ROC-AUC of 0.73 (95% CI, 0.71–0.74) (Figure 1B,D and Figure 2A). The most informative SNPs were: *SOD2* rs4880, *MGST3* rs9333378, intergenic rs4389005, *ABCA10* rs10491178, *ABCA12* rs10498027, *MCM8* rs3761873 (Table 3). Out of 30 models, these SNPs were selected 15, 9, 7, 6, 5, and 4 times, respectively (Appendix A). Only the intergenic rs4389005 has been pre-selected from the cross-validated GWAS. All other SNPs had a *p* value > 1 × 10^−4^ and were pre-selected from a systematic review of genes shown to be related with cisplatin metabolism or ototoxicity. The two most influential SNPs according to SHAP metrics [30] were *SOD2* rs4880 and *MGST3* rs9333378 (Figure 3). Homozygous genotypes for the risk alleles *SOD2* rs4880:AA and *MGST3* rs9333378:AA were found in 47% of patients who replied FACT/GOG-Ntx6 = 0 or 1, 63% of patients who replied FACT/GOG-Ntx6 = 2, and 76% of patients who replied FACT/GOG-Ntx6 = 3 or 4 (chi-squared *p* value 1 × 10^−4^).

For each sample, prediction scores ranged between 0 and 1, where a value closer to 1 indicated a higher probability of hearing loss. Using a default cut-off of 0.50, a sensitivity of 67% was reached and a positive predictive value of 51%. Correspondingly, this resulted in a specificity of 66% and a negative predictive value of 80% (Figure 2B). The model performed best on patients with the highest toxicity (Figure 2C).

For most patients (320 out of 393), adding genetic data improved hearing loss prediction; however, for 42 out of 320, this was still not enough to correctly classify these patients. In 7 out of 393 patients, the addition of genetic data led to misclassification. For 55 out of 393 patients, neither clinical nor genetic data helped on prediction and/or classification (Appendix A).

Overall, we were able to improve prediction performance when adding genetic features to clinical data (ROC-AUC 0.73) compared to the models with only clinical data (ROC-AUC 0.66).

To test robustness and non-randomness of the selected models, in the models with only clinical data, all variables were permutated, which led to a mean ROC-AUC close to 0.50 throughout the forward feature selection (Appendix A). The mean ROC-AUC for the random model with two features was 0.50 (95% CI, 0.49–0.51) (Appendix A).

In an additional test, random genetic variants were added to the model with the informative clinical features (age at diagnosis and number of treatment cycles). Mean ROC-AUC was 0.67 (95% CI, 0.66–0.68) for the model with six random genetic variants and two informative clinical features (Appendix A), which was not so different from the ROC-AUC with two clinical features only (ROC-AUC of 0.66 (95% CI, 0.65–0.66)), indicating that the random genetic variants were indeed not adding any relevant information for the prediction. From this point on, ROC-AUC started to steadily decrease as more randomly selected SNPs were added to the models (Appendix A).

## 4. Discussion

In this study, we present a model for the prediction of hearing loss after cisplatin-containing chemotherapy based on a combination of clinical and genetic features, achieving a classification performance of ROC-AUC 0.73. We observed an improved prediction after the inclusion of genetic data compared to clinical data only. Age at diagnosis and number of treatment cycles were the most important clinical predictors, matching what has previously been reported [7,9,31].

We have focused on hearing loss as part of ototoxicity, as we did not observe a strong correlation between hearing loss (FACT/GOG-Ntx6), and tinnitus (FACT/GOG-Ntx7), which may indicate independent biological mechanisms. Indeed, not all people who suffer from hearing loss have tinnitus, and vice versa, and studies on the genetics behind tinnitus are still at an early stage [32,33].

The first SNP selected in the model, the functional rs4880 SNP, is located on codon 16 exon 2 of *SOD2* that codes for the superoxide dismutase 2 (SOD2), a mitochondrial protein [34]. SNP rs4880 is the most studied *SOD2* SNP [35]; however, there is no agreement regarding how it influences SOD2 enzymatic activity. SNP rs4880 (A > G, Val16Ala) is predicted to change the structure of the SOD2 mitochondrial targeting sequence, converting a beta-sheet secondary structural motif to a partial alpha-helix [36]. Some state that due to partial arrest of the beta-sheet structure during transport across the inner mitochondrial membrane, this will likely inhibit efficient mitochondrial import of Val-SOD2 precursors and, thus, reduce enzyme activity [37]. A follow-up study has reported that the Ala variant, associated with increased SOD2 activity according to the previously mentioned study, was associated with hearing damage in cisplatin-treated pediatric medulloblastoma [38]. However, others have measured SOD2 activity and observed that it was lower in *SOD2* rs4880:GG individuals compared with *SOD2* rs4880:AA, or *SOD2* rs4880:GA [39].

The second SNP selected in the model, SNP rs9333378, is located in *MGST3*, that codes for the microsomal glutathione S-transferase 3 (MGST3) [34]. Among the microsomal glutathione S-transferases, MGST1, MGST2, and MGST3 have been reported to be important in the detoxification process [40].

Here, we hypothesized a combined effect of *SOD2* rs4880 and *MGST3* rs9333378 on cisplatin-induced hearing loss.

When platinum enters the cells, it is metabolized by the mitochondria, which will lead to the production of reactive oxygen species (ROS) such as superoxide. SOD2 will then degrade superoxide into hydrogen peroxide until complete superoxide anion degradation. If SOD2 is prevented from entering the mitochondria due to partial arrest of beta-helix, this may lead to an accumulation of ROS. ROS cause lipid peroxidation, activation of pro-inflammatory factors, and cell death by apoptosis, including hair cells [41,42]. Indeed, we observed the A-allele with a higher frequency in patients who reported hearing loss (odds ratio = 1.55, 95% CI: 1.13–2.13), contrary to what has been reported previously [38]. Furthermore, glutathiones, including glutathione S-transferase, are known to help with complete superoxide anion degradation [38]. The Genotype-Tissue Expression (GTEx) database [43] reports lower MGST3 expression levels for rs9333378:AA compared to the rs9333378:GG genotype in the brain. It is hypothesized that the rs9333378 variant leads to accumulation of cisplatin in the hair cells through decreased MGST3 activity.

Additionally, potential novel variants associated with cisplatin-induced hearing loss were selected on the logistic regression model. SNP rs4389005 located in an intergenic region was found in the cross-validated GWAS. The closest gene is *GPR12* (64 kilo base pairs 5′ to canonical transcription start site), a G protein-coupled receptor (GPCR). GPCRs have been seen to be involved in several physiological and pathological functions [44]. The subsequent SNPs, found via systematic review search, and with contribution to model performance, were SNPs *ABCA10* rs10491178 and *ABCA12* rs10498027, both leading to stop-gains within the ABCA genes which encode ATP-binding cassette (ABC) transporters. Overexpression of ABC transporters have been associated with multidrug resistance, including cisplatin, in multiple tumors [45]. *ABCA10* rs10491178:GG has been associated with lower expression of ABCA6 [43], which can lead to higher sensitivity to cisplatin and higher toxicity [46]. *MCM8* rs3761873 was the last SNP selected by the model, which leads to a stop-gain. *MCM8* encodes the mini-chromosome maintenance 8 homologous recombinant repair factor protein (MCM8), and in a recent mouse study, inhibition of MCM8 (and MCM9) hypersensitized cells to cisplatin [47].

While we observed a false discovery rate of 49% using a 0.50 cut-off, it is promising to see that only four of the twenty-three patients with the highest score (FACT/GOG-Ntx6 = 4) were misclassified. Three of them had a prediction score very close to 0.50 (two patients with 0.48 and one with 0.49 prediction scores). The last misclassified patient had a prediction score of 0.31 and was also the youngest of the 23. Furthermore, he received one of the lowest amounts of cisplatin (300 mg/m^2^ and three treatment cycles). This points to other relevant features that led this patient to develop hearing loss, either clinical or genetic predispositions that might be underrepresented in this dataset and, hence, may not have been detected.

The diagnosis of hearing loss is challenging to perform, and its definition is still far from being robustly defined [48]. Here, several potential factors for hearing loss were not explored, such as noise, infection, or vascular problems, and the toxicity was assessed several years after exposure. However, long-term toxicity also has a high impact on quality of life [9] and may be important to predict for balancing treatment intensity.

The models were trained on labels that derive from the FACT/GOG-Ntx questionnaire, which are not objectively quantified. Other measurements, such as pure-tone audiometry, which are not yet implemented routinely in clinical practice, could have been undertaken to improve precision [48]. On the other hand, using quality of life measures ensures that the focus is on the patient [49]. For instance, objective measurements might detect a similar level of toxicity between two individuals; however, only one may be affected by the symptoms and, thus, objective measurements may not be a true assessment of quality of life.

Further, BMI, as well as information about alcohol consumption and smoking habits, were retrieved in 2014 when the questionnaire was completed. These clinical features were used as a proxy at the time of treatment, but they may not represent the true values. While those features were not selected in the final model, we are unaware if the real values at the time of treatment could have added relevant information to the model. Incorporating longitudinal data, such as information collected one year after treatment, could also have been advantageous in further improving the model’s performance.

Finally, models in this study were trained on 393 patients adhering to most of the best practices of healthcare-related prediction models [50] using a logistic regression with cross-validated GWAS; nonetheless a future replication on a larger and independent patient cohort would be warranted.

Cisplatin is essential in treatment of several neoplasms; however, the inability to predict how patients will react to chemotherapy represents a major challenge, and hearing loss is one of the most common late side effects of cisplatin-based chemotherapy. In this study, we present a logistic regression with cross-validated GWAS prediction model based on a combination of genetic and clinical features able to classify patients at high (67% sensitivity), or low risk (66% specificity) of hearing loss after cisplatin-based treatment.

We also propose a combined effect involving *SOD2* rs4880 and *MGST3* rs9333378 on cisplatin-induced hearing loss development. In our study, these SNPs have not yielded significant results when single associations between SNPs and outcome have been performed; thus, a combination of cross-validated GWAS and systemic review search is suggested as a feature selection approach for machine learning.

Following confirmation in a prospective clinical setting and replication in larger independent studies, such a model could be used as a complement to support clinical decision-making and help in reducing hearing loss cases by adjusting treatment for patients in the high-risk group, especially with treatment of other cancers where cisplatin is used.

## Figures and Tables

**Figure 1 cancers-15-03923-f001:**
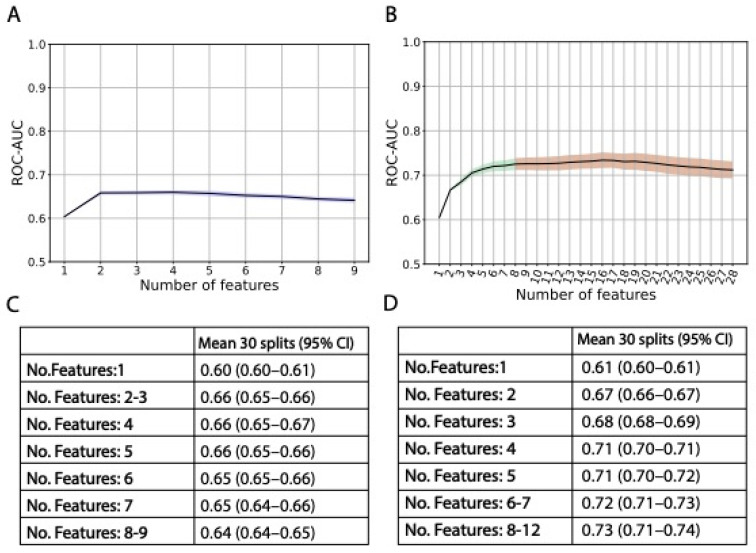
ROC-AUC mean (30 random data splits) performances in each step of the forward feature selection. (**A**): Model with clinical data with forward feature selection up until nine features. Shaded blue area indicates 95% CI. Exact ROC-AUC mean and 95% CI in (**C**). (**B**): Model with clinical and genetic data with forward feature selection up until 28 features. Shaded areas indicate 95% CI, blue color indicates that only clinical data were added, green color that clinical and genetic data were added, and red color that ROC-AUC reached a plateau. Exact ROC-AUC mean and 95% CI in (**D**). For illustration purposes, exact ROC-AUC mean and 95% CI are not indicated in (**D**) from 13 features. From 13 to 28 features, ROC-AUC mean (95% CI) was 0.73 (0.71–0.75) (13–15 features); 0.73 (0.71–0.75) (14–15 features); 0.73 (0.72–0.75) (16–17 features); 0.73 (0.71–0.75) (18–21 features); 0.72 (0.70–0.74) (22–25 features); 0.72 (0.70–0.73) (26 features); and 0.71 (0.69–0.73) (27–28 features). ROC-AUC = area under the receiver operating characteristic curve; No. = number; CI = confidence interval.

**Figure 2 cancers-15-03923-f002:**
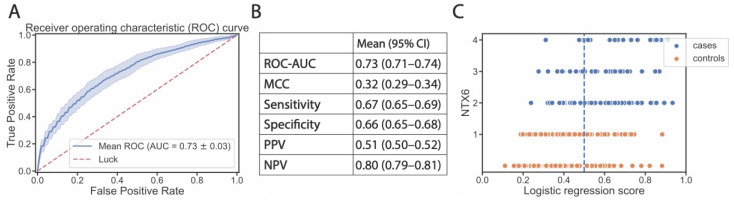
Final model performance measures and prediction scores. (**A**): Model performance shown as ROC-AUC curve. Solid blue line and shaded area indicate the mean and standard deviation across 30 data splits. Dashed red line indicates a random classifier. (**B**): ROC-AUC and other performance measures, i.e., MCC, sensitivity, specificity, PPV, and NPV using a cut-off of 0.50. (**C**): Final prediction scores (*x*-axis) for each patient, represented by a dot. Orange dots represent controls or non-affected patients (FACT/GOG-Ntx6 score 0–1), while blue dots represent cases or affected patients (FACT/GOG-Ntx6 score 2–4). Dashed vertical line represents a cut-off of 0.50, where patients with a prediction score of 0.50 or higher are considered cases. ROC-AUC = area under the receiver operating characteristic curve; MCC = Matthews correlation coefficient; PPV = positive predictive value; NPV = negative predictive value.

**Figure 3 cancers-15-03923-f003:**
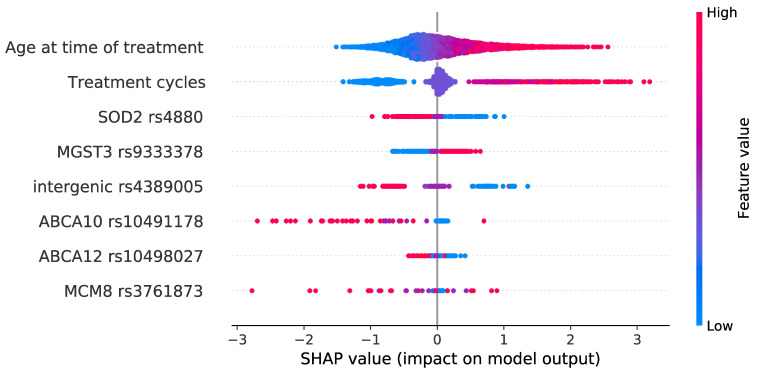
SHAP value feature importance. Individual features are ranked by importance, where age at diagnosis is the most important feature. The color represents the feature value (red: high; blue: low). Negative SHAP values (*x*-axis) contribute toward a negative model outcome (control or non-affected), while positive SHAP values contribute toward a positive model outcome (case or affected).

**Table 1 cancers-15-03923-t001:** Overview of the gene database search identifying genetic markers associated with cisplatin metabolism and ototoxicity.

		Description	Number of Genes
Cisplatin metabolism	Resistance Pathway (KEGG Pathways)	Overview of genes and interactions resulting in platinum-based drugs resistance.	46
	Detoxification Pathway (BioCyc Pathway)	Cisplatin is degraded via the glutathione-mediated detoxification pathway.	9
	Glutathione Transferases Cytochrome P450 Enzymes ABC Transporters (Uniprot)	The three protein groups may be associated with cisplatin-introduced neurotoxicity, since they affect the uptake and disposition. Genes associated with the groups were identified with Uniprot.	26 61 49
	Cisplatin (Uniprot)	Systematic search identifying cisplatin-related genes conducted with Uniprot.	22
	Cisplatin (DrugBank)	The DrugBank database contains information on pharmaceutical drugs including cisplatin.	31
Ototoxicity	Sensorineural Hearing Loss	Sensorineural hearing loss-related genes conducted with Uniprot.	155
	Ototoxicity	Ototoxicity-related genes conducted with Uniprot.	2

**Table 2 cancers-15-03923-t002:** Comparison of baseline characteristics between affected (FACT/GOG-Ntx6 score 2–4) and non-affected (FACT/GOG-Ntx6 score 0, 1) patients. Out of 433, 9 patients had not replied on FACT/GOG-Ntx6; thus, only 424 patients are represented in the table. Values show the median and interquartile range (IQR; 25–75%), or number of patients and percentages (%). IQR = interquartile range; BMI = body mass index; BEP = bleomycin–etoposide–cisplatin; GFR = glomerular filtration rate.

		Affected, Number (%)	Non-Affected, Number (%)	*p* Values ^a^
Number of patients		146 (34.4)	278 (65.6)	-
Age at diagnosis, median (IQR)		34 (27–41)	29 (26–36)	0.002
BMI, median (IQR) Unknown: 8 Affected; 10 Non-affected		21 (19–27)	22 (19–26)	0.38
BEP regimen	Normal dose	113 (78.5)	260 (95.6)	6 × 10^−27^
	Double dose	31 (21.5)	12 (4.4)	
	Unknown	2	6	
GFR before treatment, median (IQR), mL/min/1.73 m^2^ Unknown: 2 Non-affected		122 (111–135)	121 (110–133)	0.68
Cisplatin, median (IQR), mg/m^2^		400 (385–403)	400 (300–400)	*p* < 0.001
Treatment cycles	3	30 (20.5)	86 (30.9)	1 × 10^−5^
	4	85 (58.2)	180 (64.7)	
	5 or more	9 (6.2)	10 (3.6)	
	High-dose	22 (15.1)	2 (0.7)	
Histology	Seminoma	34 (23.3)	54 (19.4)	0.42
	Non-Seminoma	112 (76.7)	224 (80.6)	
Prognostic group	Good	103 (70.5)	239 (86)	*p* < 0.001
	Intermediate	32 (21.9)	30 (10.8)	
	Poor	11 (7.5)	9 (3.2)	
Alcohol consumption in number of units per week		5 (1–10)	5 (2–10)	0.30
smoking	Never	61 (41.8)	128 (46.4)	0.40
	Former	55 (37.7)	88 (31.9)	
	Current	30 (20.5)	60 (21.7)	
	Unknown	-	2	

^a^ *p* values were calculated by 2-sided Mann–Whitney *U* test for continuous or ordinal characteristics. For “histology”, *p* value was calculated by χ^2^ test. All tests are appropriate for unpaired data, and in the case of continuous variables, non-normal distributed data. Distribution of continuous variables was assessed through Shapiro–Wilk normality test.

**Table 3 cancers-15-03923-t003:** Single nucleotide polymorphisms (SNPs) selected on the final prediction model. SNPs are ordered by genetic position, and not by the number of times selected in the model. Chr. = Chromosome; SNP = single nucleotide polymorphism; MAF = minor allele frequency; CEU = European; OR = odds ratio; CI = confidence interval.

Chr.	SNP	Genetic Position ^a^	Gene	Reference Allele	Alternative Allele	Risk Allele	MAF (CEU)	Effect	OR (95% CI) ^b^	*p* Values ^c^
1	rs9333378	165601466	MGST3	G	A	A	G: 0.39	Splice acceptor	1.37 (1–1.86)	0.0441
2	rs10498027	215820013	ABCA12	G	A	G	A: 0.40	Stop gained	1.11 (0.81–1.51)	0.5158
6	rs4880	160113872	SOD2	A	G	A	G: 0.47	Missense	1.55 (1.13–2.13)	0.007183
13	rs4389005	27399338	GPRR12 (nearest gene)	A	G	A	A: 0.45	Intergenic	2.09 (1.56–2.89)	7 × 10^−6^
17	rs10491178	67149973	ABCA10	G	A	G	A: 0.07	Stop gained	1.84 (0.80–4.22)	0.1525
20	rs3761873	5939214	MCM8	A	C	A	C: 0.06	Stop gained	1.35 (0.66–2.75)	0.4148

^a^ Genetic position based on NCBI Human Genome Build 37 coordinates. ^b^ Odds ratio with 95% confidence interval for the risk allele. ^c^ A logistic model was adjusted for cisplatin dosage and age at completion of the questionnaire, and *p* values represent how likely the variant association was by random chance.

## Data Availability

For data transparency, the GWAS summary statistics, on the entire cohort are available through the NHGRI-EBI GWAS Catalog under study accession number GCST90133383. Note that performance in the machine learning models is higher than the single locus association tests represented in GWAS summary statistics, as machine learning capitalizes on non-linear correlations between individual SNPs as well as with other patient backgrounds and clinical information. In order to train and evaluate the model described in this work, SciKit-learn [28] (v0.23.2) in Python (v3.6.10) was used. PLINK [29] (v1.9) was used to perform the logistic regression association on the training set of the inner fold. The code used in this study is available at https://github.com/sblg/LR_cross_validated_GWAS, accessed on 12 November 2022. Further details and other data that support the findings of this study are available from the corresponding author upon request.

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
