# Peer review of "Predicting Hearing Loss in Testicular Cancer Patients after Cisplatin-Based Chemotherapy"

_cancers, 2023, doi:10.3390/cancers15153923_

Round 1
Reviewer 1 Report
I read the manuscript with great interest.
Well structured, sound statistical methodology, very interesting and comprehensible.
The article is publishable as is.
Author Response
No comments to be addressed.
Reviewer 2 Report
Title: Critical Review of "Predicting Hearing Loss in Testicular Cancer Patients after Cisplatin-Based Chemotherapy"
In the submitted manuscript, the authors aim to predict hearing loss in testicular cancer patients after cisplatin-based chemotherapy. The topic studied and the potential implications for patient care are significant. However, there are several areas in the manuscript that require further attention and clarification. My main concerns regarding the manuscript are as follows:
1. Introduction and Background: The manuscript lacks a comprehensive introduction that provides sufficient context for the study. It is essential to provide an overview of the prevalence and severity of hearing loss in testicular cancer patients receiving cisplatin-based chemotherapy. Additionally, a thorough discussion of the existing literature on risk factors, mechanisms, and predictive models for hearing loss in this population would be beneficial to establish the novelty and significance of the proposed research.
2. Methodology and Data Collection: The manuscript does not adequately describe the methodology and data collection process. It is crucial to provide detailed information on how the patient cohort was selected, the chemotherapy protocols used, and the specific audiometric tests employed to assess hearing loss. Additionally, it is essential to describe any control groups or comparative analyses that were conducted to validate the predictive model.
3. Statistical Analysis and Model Development: The manuscript lacks a clear description of the statistical analysis performed and the methodology employed to develop the predictive model. It is essential to provide information on the variables included in the model, the model building process (e.g., feature selection methods, model validation techniques), and the performance metrics used to evaluate the predictive accuracy of the model. Without these details, it is challenging to assess the robustness and reliability of the proposed predictive model.
4. Validation and Generalizability: The manuscript does not discuss the validation of the predictive model. It is crucial to validate the model using independent datasets to assess its generalizability and predictive performance in different patient populations. Furthermore, a discussion on the limitations and potential sources of bias in the study design should be included to provide a balanced interpretation of the findings.
5. Clinical Relevance: The manuscript should discuss the potential clinical implications of the predictive model. How can this model aid in clinical decision-making? Are there any interventions or preventive measures that can be taken based on the predicted risk of hearing loss? Providing a clear discussion of the clinical utility of the predictive model will enhance the significance and relevance of the study.
6. Future Directions: The manuscript would benefit from a section dedicated to discussing future directions for research in this area. Are there any potential improvements or refinements that can be made to the predictive model? Are there other relevant factors or biomarkers that could be incorporated to enhance the predictive accuracy? Addressing these points will contribute to the ongoing scientific discourse and potential advancements in the field.
Overall, addressing these concerns will strengthen the manuscript, ensuring a more comprehensive and rigorous analysis of the predictive model for hearing loss in testicular cancer patients receiving cisplatin-based chemotherapy.
The quality of English language in the manuscript is generally good. The authors demonstrate a proficient command of the language, and the writing is clear and understandable.
Round 2
Reviewer 2 Report
The authors have made satisfactory changes in this revision. I am satisfied with the current state and would like to recommend publication.